# Self-Concept and Inattention or Hyperactivity–Impulsivity Symptomatology: The Role of Anxiety

**DOI:** 10.3390/brainsci10040250

**Published:** 2020-04-23

**Authors:** Marisol Cueli, Celestino Rodríguez, Laura M. Cañamero, José Carlos Núñez, Paloma González-Castro

**Affiliations:** Department of Psychology, University of Oviedo, 33003 Asturias, Spain; cuelimarisol@uniovi.es (M.C.); lauramcanamero@uniovi.es (L.M.C.); jcarlosn@uniovi.es (J.C.N.); mgcastro@uniovi.es (P.G.-C.)

**Keywords:** ADHD, ADHD-presentations, self-concept, anxiety, comorbidity

## Abstract

Attention-deficit/hyperactivity disorder (ADHD) has been associated with low levels of self-concept (academic, emotional, social or physical), although this association can differ in the function of the inattention or hyperactivity–impulsivity symptomatology. Furthermore, the relation between ADHD and self-concept can be mediated or moderated by the levels of anxiety. This work is aimed to examine the differential effect of inattention symptomatology and hyperactivity–impulsivity symptomatology on academic, emotional, social and physical self-concept and the mediating or moderating role of anxiety in this relationship. A total of 167 students (70.7% boys and 29.3% girls) aged between 11 and 16 participated in this study. Students’ ADHD symptomatology, self-concept in four areas (academic, emotional, social and physical self-concept) and trait anxiety were measured with the State-Trait Anxiety Inventory for Children. The results indicate that trait anxiety mediates the relationship between inattention and emotional, social and physical self-concept but does not moderate this relationship. Trait anxiety does not mediate or moderate the relationship between hyperactivity–impulsivity symptoms and self-concept. When inattention symptomatology increases, academic self-concept decreases directly, but students’ emotional, social and physical self-concept decreases indirectly through trait anxiety.

## 1. Introduction

Attention-deficit/hyperactivity disorder (ADHD) is a persistent pattern of inattentive, restless and impulsive behavior that is more frequent and severe than that typically observed in subjects at a similar stage of development [1]. The classification of the Diagnostic Statistical Manual (DSM) in its fifth edition [1] includes ADHD as a neurodevelopmental disorder and differentiates between three types of presentations (predominantly hyperactive/impulsive, predominantly inattentive and a combined presentation). The prevalence of this disorder is estimated to be 5.9–7.1% in childhood and adolescence and 5% in adults [2]. 

Childhood ADHD has been associated with impairment in academic achievement, family interaction, peer relationships, self-esteem, and quality of life [3]. Furthermore, ADHD is commonly comorbid with externalizing and internalizing disorders, such as learning difficulties, depression, oppositional defiant disorder, behavior disorders, anxiety and mood disorders [4,5,6]. Comorbidity with anxiety is estimated at 20–25% [4,7] with higher rates of agoraphobia, simple phobias, separation anxiety, social phobia, and obsessive-compulsive disorder [8]. 

ADHD with comorbid anxiety is associated with lower self-esteem and more stressful life events [9,10]. For example, Castagna et al. [11] indicated that young people diagnosed with an ADHD combined presentation, had more frequent personal failure and hostile intent negative self-statements in comparison with those diagnosed with an inattentive presentation. In study [11], authors also showed that the association of ADHD presentation and negative self-statements was moderated by anxiety; specifically, negative self-statements of personal failure were highest in children with anxious and ADHD combined presentation. 

According to the multidimensional model of self-esteem [12], children who frequently experience failure are at risk of developing a lower sense of self-concept. Conversely, children who often experience success may develop an enhanced sense of self-concept. ADHD is associated with poor grades, increased rates of detention and expulsion, and low rates of high school graduation. This group of children are also likely to experience more negative social events, such as being rejected by peers, having poor social skills. Thus, according to Kita and Inoue [13], these negative experiences can have a large impact on self-esteem and self-concept in children with ADHD. For this reason, some studies have focused on self-esteem and self-concept in children with ADHD as an important topic affecting individuals with this disorder [3,14]. 

According to Harpin et al. [3], more than half of the previous studies found that children with ADHD had lower self-esteem compared with healthy controls. However, other studies suggest that some children with ADHD rate their quality of life as being less negative compared with evaluations made by their parents [14] and some children with ADHD even tend to overestimate their own competence [15,16]. For example, Jia et al. [16] highlighted that despite the presence of difficulties, children with ADHD tend to hold overly positive self-perceptions of their competence, a phenomenon referred to as Positive Bias (although children with ADHD usually have lower social and behavioral competence, they can tend to report equivalent or more favorable self-assessments of their own competence, which has been associated with risky driving behavior, behavioral challenges and social deficits, and poorer response to treatment [17,18]). 

Taking into account these differences between the previous studies, it is unclear whether children with ADHD have low self-esteem and self-concept during childhood owing to negative life experiences. Self-esteem is related to the assessment that the person makes about themselves, expressing approval or disapproval and indicates how capable, important and valuable they consider themselves [19]. Joined to the concept of self-esteem, self-concept is defined as a multidimensional assumption resulting from the interaction with the environments which children or adults are involved in during the process of social construction [20]. Given that the self-concept is multidimensional, it can be assessed in academic, emotional, social or peer relations, and physical areas [13,21]. 

Research has indicated that a child’s self-concept goes through a major transition at the age of 8 or 9 [22]. Although the self-concept of a very young child is generally quite high, that may change as the child ages and has experiences that reveal individual strengths and weaknesses [22]. Self-concept and self-esteem have been suggested to be internalized during the same developmental period as when ADHD is generally diagnosed and treated [23]. 

Bussing et al. [23] used the Pier-Harris Self-Concept Scale in a study designed to examine how levels of self-concept may be affected by ADHD characteristics. The results of the study showed that children with ADHD exhibited lower scores of total self-concept than their peers who did not meet the criteria for diagnosis. Furthermore, they showed that scores were lower for children with ADHD who had comorbid internalizing symptoms of depression or anxiety. In particular, those students with ADHD and internalizing symptoms scored worst in the areas of anxiety (emotional self-concept) and popularity (social self-concept). However, in this study, students were younger and at the age where self-concept is still developing. 

Moreover, Kita and Inoue [13] carried out a study with 564 early adolescents (aged 12 to 15). They measured scholastic self-concept, athletic self-concept, behavioral conduct, social acceptance, and physical appearance. Their results indicated that ADHD in early adolescents was associated with low levels of self-concept. Specifically, the authors found that severe inattentive symptoms were associated with decreased scholastic and athletic self-concept. However, hyperactive–impulsive symptoms were associated with the lowest self-concept in terms of behavioral conduct. This study did not consider the possible role of anxiety in this relationship but the authors pointed to differences in self-concept depending on the kind of symptomatology (inattention or hyperactivity–impulsivity) [13]. In addition, Kita and Inoue [13] observed differences between boys and girls (boys scored higher with respect to scholastic self-concept, athletic self-concept, and physical appearance). 

In this sense, gender can be another relevant factor in the association between ADHD and anxiety. Gershon [24] and Rucklidge [25] found that girls with ADHD manifested more internalizing problems, with higher rates of depression and anxiety, compared to boys with ADHD. This association can have a differential effect in the self-concept of boys and girls with ADHD, although other studies did not find differences based on gender. Houck et al. [26] carried out a study with 145 children and adolescents with ADHD using the Piers–Harris Children’s Self-Concept Scale. Their results showed that more internalizing behavior problems predicted lower self-concept. However, gender did not predict self-concept in this study. The authors concluded that these relationships between gender and self-concept are not well understood and require more research [26].

Based on the results of previous research, this study aims to analyze the relationship between self-concept (assessed by a Spanish version of the Piers-Harris) and the inattentive and hyperactivity–impulsivity symptomatology of ADHD, considering the role of anxiety. To be more precise, this study examines the effect of inattention symptomatology and hyperactivity–impulsivity symptomatology on academic, emotional, social and physical self-concept and the role of anxiety as a mediator or moderator of this relationship. Given the results from Castagna et al. [11], we expect inattention symptomatology and hyperactivity symptomatology to have a strong, differential relationship with self-concept, and trait anxiety to be a mediator or moderator of this relationship. To achieve this objective, we worked with a broad sample of teenagers aged between 11 and 16, as younger children are in the process of developing their self-concept [22,23]. 

It is important to note that, in this study, the focus is on the symptomatology rather than the diagnosis. Increasing evidence suggests that ADHD stands at the end of a continuum and that ADHD symptoms may occur in the absence of the full disorder [27]. 

Finally, given the differences between girls and boys in levels of self-concept and anxiety, in this study of the hypothesis of mediation or moderation, we include gender in the corresponding model analysis [24,25].

## 2. Materials and Methods

### 2.1. Participants

In this study, 167 children participated, 118 boys (70.7%) and 49 girls (29.3%), aged between 11 and 16 years old (*M* = 13.77, *SD* = 1.24). All the children had an Intelligence Quotient (IQ) of 80 or more (*M* = 97.54, *SD* = 10.32), assessed by the Wechsler Intelligence Scale for Children-Revised [28]. Participants attended public and independent schools in the Autonomous Community of the Principality of Asturias (Spain).

The entire sample was recruited from Clinical Centers in northern Spain where the participants had been referred for a diagnosis given their symptoms of ADHD. Once informed by the researchers about the objectives and the requirements of this study, interested Clinical Centers indicated their agreement to collaborate. 

There were statistical differences in the gender-distribution of boys and girls in the sample *χ*^2^(1) = 28.509, *p* ≤ 0.001. This variable was used as a covariable in the subsequent analysis. Taking into account the results in the State-Trait Anxiety Inventory for Children [29], 53.3% of the sample reached low levels of anxiety (percentiles below 50), 27.5% of the sample reached medium levels of anxiety (percentiles between 50 and 75) and 29.2% reached high levels of anxiety (percentiles above 80). 

### 2.2. Instruments

The Scale for the Assessment of ADHD (EDAH) [30] was administered to the subjects’ families. It comprises 20 items that provide information on the presence of symptomatology related to attention deficit (5 items), hyperactivity–impulsivity (5 items) and conduct disorder (10 items). The scale helps differentiate between predominantly inattentive, predominantly hyperactivity–impulsivity and combined ADHD. In this study, the following variables were used: hyperactivity–impulsivity (the percentile score in the hyperactivity–impulsivity items) and inattention (the percentile score in the items that measure attention deficit). Higher scores indicate more probability of the presence of hyperactivity–impulsivity or attention deficit. In the original sample, Cronbach’s alpha for the total scale was 0.929. The reliability of the instrument, using Cronbach’s Alpha, was 0.77 in the current sample.

The Piers-Harris Self-Concept Scale (PH-A) [21] was completed by the participants of the study. In the Spanish version the questionnaire is designed for use with children and adolescents aged 7–18 years old. It is composed of 72 items organized by self-concept in various areas, academic (15 items; e.g., “aprendo rápido en la mayoría de las asignaturas” I learn quickly in most of my subjects), emotional (22 items; i.e., “soy una persona feliz” I am a happy person), social (22 items; “me resulta difícil encontrar amigos” I find it difficult to make friends) and physical (13 items; i.e., “soy muy ágil” I’m very nimble). The children respond to each statement “yes”, “no” or “sometimes”. Higher scores reflect a more positive self-concept (maximum score is 30 for academic self-concept, 44 for emotional, 44 for social, and 26 for physical). In the original version test–retest reliability estimates range from 0.71 to 0.96 [21] and internal consistency from 0.78 to 0.93. The reliability of the instrument, using Cronbach’s Alpha, was 0.72 in the current sample.

State-Trait Anxiety Inventory for Children (STAI-C) [29] consists of two 20-item scales that measure State and Trait anxiety in children between 8 and 14 years old. The State anxiety is usually associated to specific (i.e., “me siento descansado” I feel rested). The Trait anxiety assesses how the child generally feels (i.e., “me decido fácilmente” I make decisions easily). Children must respond in a 3-point Likert scale (State scale: not at all, somewhat, very much so; Trait scale: almost never, sometimes or often). A separate score is provided for the state scale and the trait scale to determine which type of anxiety is predominant. The variable examined in this study was trait anxiety (raw score). To interpret the information provided by STAI-C correctly, higher scores correspond to higher levels of anxiety, and vice versa (the maximum score in the Trait scale is 60). The authors report alpha coefficients of 0.90 to 0.93 for internal consistency and 0.73 to 0.86 for test-retest reliability (Cronbach’s Alpha in the current sample, was 0.58).

### 2.3. Procedure

The study obtained previous approval by the Ethical Committee of the Principality of Asturias (code: proyect 70/19), and all instructions from the protocol were performed according to institutional guidelines and laws.

Given the objective of this research, we studied participants who had been referred to Clinical Centers in northern Spain for a diagnosis given their symptoms of ADHD. Once parental consent to evaluate the children was given, the corresponding tests were conducted to participate in this study.

The study was conducted in accordance with The Code of Ethics of the World Medical Association (Declaration of Helsinki), which reflects the ethical principles for research involving humans [31].

### 2.4. Design and Data Analysis

Data analyses were conducted in four steps. Firstly, the descriptive statistics for the variables being studied were analyzed, paying special attention to skewness and kurtosis. Furthermore, two multivariate analysis of variances (MANOVAS) were carried out in order to analyze gender differences in the four self-concept variables and the anxiety variable. 

Secondly, we performed an analysis of the mediating role of trait anxiety in the effect of inattention symptomatology (independent variable) on the four academic self-concepts being assessed (academic, emotional, social and physical). Figure 1 presents the mediational model to be tested. 

In the third step, we carried out an analysis of the mediating role of trait anxiety in the effect of hyperactive–impulsive symptomatology (independent variable) on the four academic self-concepts being assessed. Figure 2 presents the mediational model to be tested. 

Finally, we performed two analyses of trait anxiety’s moderation of the effect of inattention or hyperactivity–impulsivity on the four self-concepts being assessed. 

The mediation and moderation analyses were carried out using the PROCESS model in the Statistical Package for the Social Sciences (SPSS) [32] (version 22.0). The effect size was calculated using Cohen’s *d* [33]: *d* < 0.20 = minimum effect size; 0.20 < *d* < 0.50 = small effect size; 0.50 < *d* < 0.80 = medium effect size; *d* > 0.80 = large effect size.

## 3. Results

### 3.1. Preliminary Analyses

Table 1 provides the descriptive statistics for the variables and the Pearson correlation matrix. According to the skewness and kurtosis values, univariate normality is observed in the variables of interest. Correlation analysis showed a negative relationship between the self-concept variables and trait anxiety. In addition, inattentive symptomatology correlates negatively with academic self-concept, and social self-concept. This result shows that with higher levels of inattentive symptomatology, there are lower levels of academic and social self-concept. Hyperactivity–impulsivity symptomatology does not exhibit a correlation with the self-concepts assessed or with trait anxiety. Regarding age, academic self-concept showed a negative and significant correlation, while social self-concept showed a positive and significant correlation. 

We carried out two MANOVAS with self-concept and anxiety variables as dependent variables and gender as an independent variable. The data indicate that, for self-concept, the differences between boys and girls were not statistically significant *F*(4, 162) = 2.080, *p* = 0.086, η_p_^2^ = 0.049. For anxiety, there were significant differences between boys and girls *F*(2, 164) = 3.962, *p* = 0.021, η_p_^2^ = 0.046, albeit with a minimum effect size. Gender was included in the appropriate place in the mediation and moderation models.

### 3.2. Mediation Analysis for Inattentive Symptomatology

The results of the mediation analysis for inattention symptomatology are provided in Table 2 and Figure 3. In general terms, trait anxiety mediates the relationship between inattention symptomatology and emotional, social and physical self-concept. 

Inattention symptomatology exhibits a positive, significant relationship with trait anxiety, and trait anxiety exhibits significant, negative relationships with academic, emotional, social and physical self-concept showing that with higher anxiety, there are lower levels of self-concept in these four areas. Although inattention symptomatology does not have a significant direct effect on emotional, social and physical self-concept, it does have an indirect effect through trait anxiety with large effect sizes. However, in the case of academic self-concept, inattention has a direct effect (with a large effect size) but not an indirect effect. 

Additionally, based on the data in Table 2, it was correct to include the gender variable in the simple mediation model. Gender had a significant effect on trait anxiety but not on academic, emotional, social or physical self-concept. 

### 3.3. Mediation Analysis for Hyperactive Symptomatology

The results of the mediation analysis for hyperactive symptomatology are provided in Table 3 and Figure 4. In general terms, trait anxiety does not mediate the relationship between hyperactivity symptomatology and academic, emotional, social or physical self-concept. 

Hyperactivity–impulsivity symptomatology does not demonstrate a significant relationship with trait anxiety, although trait anxiety demonstrates significant, negative relationships with academic, emotional, social and physical self-concept (with large effect sizes). In this case, hyperactivity symptomatology does not have a direct or indirect effect on the self-concepts assessed. Again, it was correct to include the gender variable in the simple mediation model (Table 3), given that it had a significant effect on trait anxiety although not on academic, emotional, social or physical self-concept.

### 3.4. Moderation Analysis

The results of the moderation analysis indicate that trait anxiety does not alter the relationship between inattention symptomatology and academic self-concept (*b* = 0.004, *p* = 0.227), emotional self-concept (*b* = 0.003, *p* = 0.411), social self-concept (*b* = −0.007, *p* = 0.185), or physical self-concept (*b* = 0.003, *p* = 0.375). Similarly, trait anxiety does not modify the relationship between hyperactivity–impulsivity symptomatology and academic self-concept (*b* = −0.002, *p* = 0.161), emotional self-concept (*b* = −0.001, *p* = 0.526), social self-concept (*b* = −0.003, *p* = 0.194), or physical self-concept (*b* = 0.000, *p* = 0.992). 

## 4. Discussion

This study aimed to examine the differential effect of inattention symptomatology and hyperactivity–impulsivity symptomatology on academic, emotional, social, and physical self-concept and the role of anxiety as a mediator or moderator of this relationship. The results showed a differential pattern in the case of inattentive symptomatology and in hyperactive symptomatology.

In terms of mediation, inattentive symptomatology has a direct effect on students’ academic self-concept but not on their emotional, social or physical self-concept, where the effect was indirect. This result indicates that when students have more inattention symptomatology, they also have worse perceptions of their ability to successfully solve school tasks. However, the presence of anxiety does not have a significant impact on this relationship between inattention and academic self-concept. In the case of emotional, social and physical self-concept, inattention has an indirect effect through anxiety levels, with a negative relationship. Students with more inattention show lower levels of emotional, social and physical self-concept. We can conclude that, in the present sample, trait anxiety mediates the relationship between inattention and emotional, social and physical self-concept. 

On the other hand, looking at the mediating effects of trait anxiety in hyperactivity–impulsivity symptomatology and self-concept, the results show that there was neither a direct nor indirect effect. There was no mediation relationship between these variables. It is necessary to highlight that, in this study, we have examined ADHD symptomology as a continuum, under the assumption that ADHD symptoms exist in the absence of the full disorder [27]. In this sense, it is possible that, considering the diagnosis or even the severity of the symptoms, the results and conclusions change showing different profiles. 

The differences between inattention and hyperactivity–impulsivity symptomatologies could be explained in relation to the positive bias that has been associated with ADHD [15,16]. Previous research has found that not all children with ADHD demonstrate this positive bias [34], and more specifically, the overestimation of competency in children with ADHD has been linked to behavior problems and aggression over time [35,36]. In this sense, it is also possible that poor estimation of competency could be more associated with hyperactive–impulsive rather than inattentive symptomatology. 

The results follow similar lines as those from Kita and Inoue [13], who observed differences in self-concept based on inattentive and hyperactive–impulsive symptomatology. They found that severe inattentive symptomatology was associated with decreased self-concept in terms of school and athletic self-concept, both of which are related to school activities and classes. Similarly, in our study, the presence of inattentive symptomatology is related to lower levels of academic self-concept in students aged between 11 and 16. However, in the study by Kita and Inoue [13], hyperactive–impulsive symptomatology was associated with the lowest self-perception in terms of behavioral conduct. In our study, we did not find any association between hyperactivity–impulsivity and self-concept, although the areas assessed were not related to personal or behavioral self-perception. 

With respect to gender, Kita and Inoue [13] saw differences between boys and girls (boys scored higher in scholastic self-concept, athletic self-concept, and physical appearance). In our sample, gender had a significant effect on trait anxiety but not on academic, emotional, social or physical self-concept.

On the other hand, this study analyzes the moderating effect of anxiety on the relationship between inattention or hyperactivity–impulsivity symptomatology and self-concept in four areas. The results indicate that anxiety does not moderate this relationship. Although the results from Castagna et al. [11], with 114 students aged between 7 and 16, showed that the association of ADHD presentation type and negative self-statements was moderated by anxiety, the same did not occur when we assessed students’ self-concept and the symptoms rather than the diagnosis. It is possible that these negative thoughts seen by Castagna et al. [11] do not condition the self-concept of students. In addition, the presence of a specific diagnosis can have a more significant impact on the students. 

In sum, although some authors have indicated that students with ADHD have fewer social skills and relationships with their peers, it seems that this not affect their social self-concept in the case of hyperactivity–impulsivity and is mediated by anxiety in the case of inattention. Authors such as Hoza et al. [37] showed that 56% of children with ADHD have no reciprocated friendships, and even when they do have friends, these friendships tend to be worse in quality and stability than those of children without the disorder. In addition, according to Hodgens et al. [38], children with ADHD have substantially lower social skills than other children, according to ratings by parents and teachers. Research has also indicated that students with ADHD often exhibit deficits in emotion regulation [39] and more internalizing problems like anxiety and depression [40]. This does not lead to the students with hyperactive–impulsive symptoms having a lower emotional self-concept; however, in the case of inattentive symptoms, the relationship with emotional self-concept is indirect. Lastly, with respect to physical self-concept, Kita and Inoue (2017) [13] found that that severe inattentive symptoms were associated with decreased athletic self-concept (physical self-concept), which chimes with our results showing that inattentive symptoms affect physical self-concept, albeit indirectly. 

Finally, we must consider some of the limitations of this study, such as the fact that the assessment of inattentive and hyperactivity–impulsivity symptoms was done using the parents’ perspective as in previous research [41]. However, in our case, the parents’ perspective may not agree with the children’s perspective. The fact that inattention and hyperactivity–impulsivity affect students’ self-concept means that children must be aware of their own problems, so the students’ perspective may produce other results and it would be interesting to compare them with our findings. Moreover, it is necessary to highlight the fact that the study has been carried out with students who had been referred to Clinical Centers given their symptomatology, but the specific diagnosis was not taken into account. Again, taking into account the diagnosis or a sample of students with moderate or severe ADHD could provide different results. Another limitation is related to the use of questionnaires and the honesty of the students. In the future, it would be interesting to introduce a sincerity scale to eliminate those students with lower levels of sincerity. Moreover, the grade of anxiety could be different among students with inattentive symptoms compared to students with hyperactivity–impulsivity symptoms. For example, within the Multimodal Treatment Study (MTA), Jensen et al. [42] showed that, in the group of children with ADHD and anxiety symptoms, inattention seems to prevail over hyperactivity–impulsivity symptoms. Also, González-Castro et al. [6] observed different levels of anxiety between ADHD presentations.

## 5. Conclusions

In our view, the results of this research contribute to the study of the emotional characteristics associated with inattention and hyperactivity–impulsivity. The findings reveal that the presence of inattention symptomatology directly or indirectly affects self-concept. Independent of anxiety, inattentive students feel less competence carrying out academic tasks. This association is expected considering that these students need more time to complete school activities, they have difficulties concentrating in class, they forget important things (material, exams, homework), and they achieve worse results in exams. However, in relation to the other self-concepts assessed, it is necessary to consider anxiety to find an association between inattention and emotional, social and physical self-concept. The results indicate that the mere presence of inattention does not have a direct effect on emotional, social or physical self-concept. Furthermore, in the case of hyperactivity–impulsivity in the present sample taking into account the symptomatology instead of the specific diagnosis, the relationship is neither direct nor indirect, and there is no association between this set of symptoms and students’ self-concept.

Considering the results, the main contribution of this article relates to the importance of considering students’ self-concept, especially those with inattentive symptomatology, paying special attention to their academic perception with the aim of avoiding them developing a negative self-concept that might affect their school performance and progress [43]. 

## Figures and Tables

**Figure 1 brainsci-10-00250-f001:**
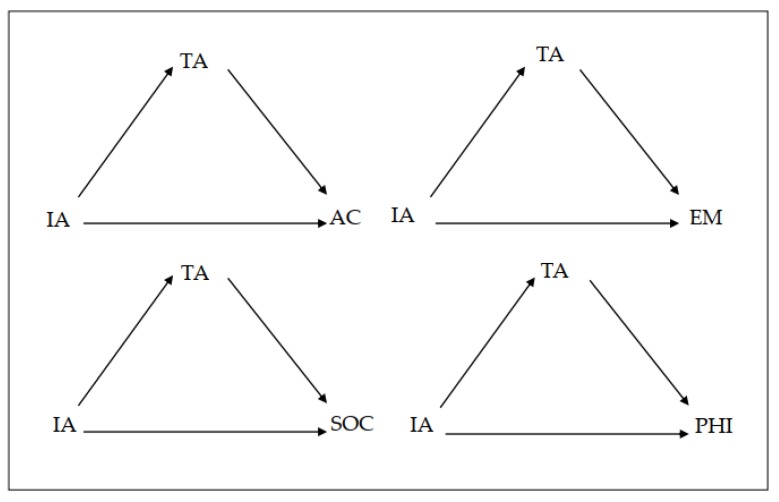
Mediational model for inattention symptomatology. AC = Academic self-concept; EM = Emotional self-concept; SOC = Social self-concept; PHI = Physical self-concept; TA = Trait anxiety; IA= Inattention symptomatology.

**Figure 2 brainsci-10-00250-f002:**
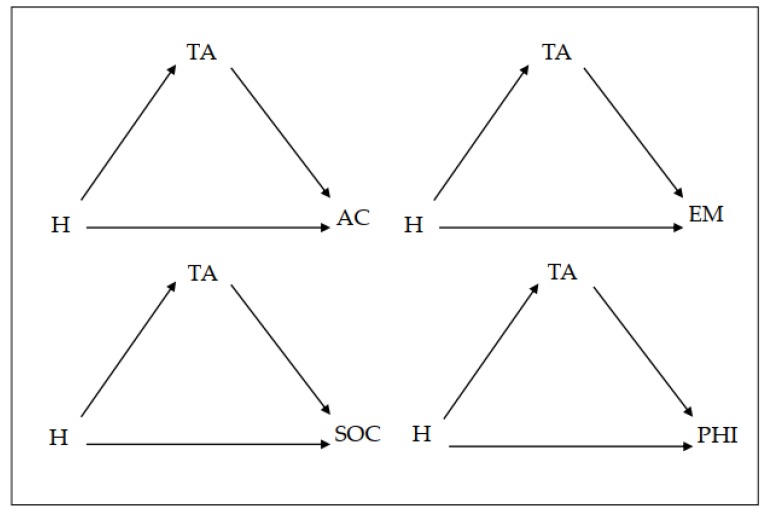
Mediational model for hyperactivity symptomatology; AC = Academic self-concept; EM = Emotional self-concept; SOC = Social self-concept; PHI = Physical self-concept; TA = Trait anxiety; IA= Inattention symptomatology.

**Figure 3 brainsci-10-00250-f003:**
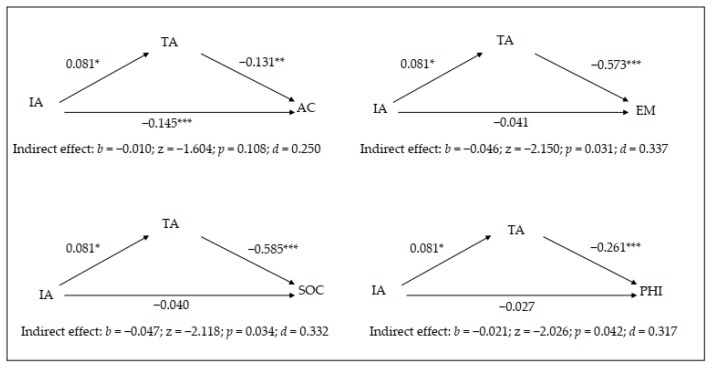
Graphical representation of the results of the mediation model for the inattention symptomatology. * *p* < 0.05; ** *p* < 0.01; *** *p* < 0.001.

**Figure 4 brainsci-10-00250-f004:**
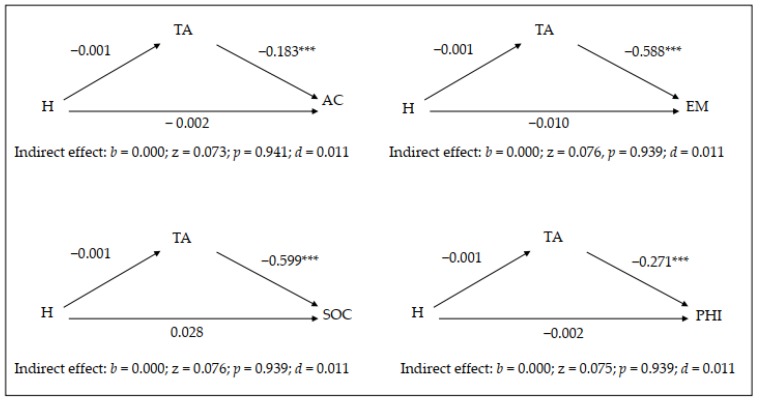
Graphical representation of the results of the mediation model for the hyperactivity–impulsivity symptomatology. *** *p* < 0.001.

**Table 1 brainsci-10-00250-t001:** Descriptive statistics and Pearson correlation matrix.

	Academic	Emotional	Social	Physical	TA	IA	H	Age
Academic	−							
Emotional	0.364 ***	−						
Social	0.181 *	0.456 ***	−					
Physical	0.314 ***	0.472 ***	0.468 ***	−				
TA	−0.250 ***	−0.613 ***	−0.516 ***	−0.423 ***	−			
IA	−0.424 ***	−0.146	−0.158 *	−0.112	0.116	−		
H	−0.011	−0.026	0.101	0.002	−0.022	0.047	−	
Age	−0.163 *	0.022	0.172 *	−0.073	−0.055	0.038	−0.073	−
M	14.24	26.00	30.32	18.56	34.89	89.28	64.78	13.77
SD	5.33	7.15	8.19	4.88	7.37	15.51	28.30	1.24
SK	0.43	−0.53	−1.07	−0.68	0.50	−3.09	−0.52	0.16
K	−0.14	−0.14	0.48	−0.15	−0.09	10.81	−0.92	−0.67
Min	4	5	7	5	20	10	5	11
Max	29	39	42	26	57	99	99	16.83

**Note.** Academic = Academic self-concept; Emotional = Emotional self-concept; Social = Social self-concept; Physical = Physical self-concept; TA = Trait anxiety; IA= Inattention symptomatology; H = Hyperactivity–impulsivity symptomatology; *M* = Mean; *SD* = Standard Deviation; SK = Skewness; K = Kurtosis; Min = Minimum; Max = Maximum. * *p* < 0.05; *** *p* < 0.001.

**Table 2 brainsci-10-00250-t002:** Results of the mediational analysis for the inattention symptomatology.

	Coefficient	SE	*t*	*p*	*d*	LLCI	ULCI
DV: Trait Anxiety							
IA	0.081	0.036	2.223	0.027	0.349	0.091	0.154
Gender	3.995	1.246	3.204	0.001	0.511	1.533	6.456
DV: Academic self−concept							
TA	−0.131	0.051	−2.541	0.012	0.401	−0.233	−0.029
IA	−0.145	0.024	−5.913	0.000	1.029	−0.194	−0.097
Gender	−1.076	0.850	−1.265	0.207	0.196	−2.756	0.603
DV: Emotional self−concept							
TA	−0.573	0.061	−9.258	0.000	2.053	−0.695	−0.451
IA	−0.041	0.029	−1.418	0.158	0.220	−0.100	0.016
Gender	−0.904	1.019	−0.887	0.376	0.137	−2.917	1.108
DV: Social self−concept							
TA	−0.585	0.076	−7.654	0.000	1.470	−0.736	−0.434
IA	−0.040	0.036	−1.102	0.271	0.171	−0.112	0.031
Gender	1.645	1.258	1.307	0.193	0.203	−0.840	4.131
DV: Physical self−concept							
TA	−0.261	0.048	−5.393	0.000	0.918	−0.357	−0.165
IA	−0.027	0.023	−1.184	0.238	0.184	−0.073	0.018
Gender	−0.964	0.798	−1.208	0.228	0.187	−2.540	0.611

**Note.** DV = Dependent Variable; TA = Trait anxiety; IA= Inattention symptomatology; M = Mean; SD = Standard Deviation; SK = Skewness; K = Kurtosis; Min = Minimum; Max = Maximum.

**Table 3 brainsci-10-00250-t003:** Results of the mediational analysis for hyperactivity symptomatology.

	Coefficient	SE	*t*	*P*	*d*	LLCI	ULCI
DV: Trait Anxiety							
H	−0.001	0.020	−0.076	0.939	0.011	−0.040	0.037
Gender	3.359	1.236	2.716	0.007	0.429	0.917	5.800
DV: Academic self−concept							
TA	−0.183	0.056	−3.273	0.001	0.523	−0.294	−0.072
H	−0.002	0.014	−0.202	0.839	0.031	−0.031	0.025
Gender	0.207	0.908	0.228	0.819	0.035	−1.586	2.001
DV: Emotional self−concept							
TA	−0.588	0.061	−9.602	0.000	2.220	−0.709	−0.467
H	−0.010	0.015	−0.689	0.491	0.106	−0.041	0.020
Gender	−0.584	0.992	−0.588	0.556	0.091	−2.543	1.375
DV: Social self−concept							
TA	−0.599	0.075	−7.975	0.000	1.568	−0.747	−0.450
H	0.028	0.019	1.484	0.139	0.231	−0.009	0.066
Gender	2.142	1.216	1.761	0.080	0.275	−0.258	4.543
DV: Physical self−concept							
TA	−0.271	0.048	−5.657	0.000	0.973	−0.366	−0.176
H	−0.002	0.012	−0.230	0.817	0.035	−0.027	0.021
Gender	−0.734	0.776	−0.945	0.345	0.146	−2.267	0.799

**Note.** DV = Dependent Variable; TA = Trait anxiety; IA= Inattention symptomatology; M = Mean; SD = Standard Deviation; SK = Skewness; K = Kurtosis; Min = Minimum; Max = Maximum.

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
