# Peer review of "Self-Concept and Inattention or Hyperactivity–Impulsivity Symptomatology: The Role of Anxiety"

_brainsci, 2020, doi:10.3390/brainsci10040250_

Round 1

Reviewer 1 Report

Review of the article entitled “Self-Concept and Inattention or Hyperactivity-Impulsivity Symptomatology: The role of Anxiety” by Dr. Cueli and coll.

brainsci-780489

This study is aimed to understand possible relationships among ADHD symptoms and self-esteem with special regard to the influence of anxiety in this context. ADHD comorbidities include emotive disorders and low self-esteem. Therefore a better understanding of reciprocal influences is warranted.

Minor revisions required

Background knowledge is extensively reported and the Introduction appears lengthy.  Previous studies reported in detail should be shortened.

The analytical plan is well done and well reported.

It is unclear to what rate and extent anxiety symptoms did occur in the study sample.

It is unclear whether there was any correlation of symptoms with participants’ age

Students with more inattention show lower levels of emotional, social and physical self-concept mediated by anxiety. It should be addressed whether anxiety levels were however higher in students with more inattention. If this is not the case, the issue should be discussed as well.

When drawing their conclusions, the Authors should pay more attention and discussion to the continuum of symptomatology in patients with ADHD. To assert that “in the case of hyperactivity-impulsivity there is no association between this set of symptoms and students’ self-concept” is a hazard in the opinion of this reviewer.

Author Response

First, we would like to thank the reviewers for their comments and for the analysis of our work. The comments have been very interesting and instrumental in helping us improve the write-up of our work. Below, you will find how we responded to each and every one of reviewer’s comments. We would be happy to respond to any other comments you may have in this new version. Please let us know if you need additional information.

  • Introduction (lines 44, 82-84, 88, 105, 118-119): We have summarized the previous studies described. [Background knowledge is extensively reported and the Introduction appears lengthy. Previous studies reported in detail should be shortened.]
  • Lines 146-149: we have included the levels of anxiety in the sample. [It is unclear to what rate and extent anxiety symptoms did occur in the study sample].
  • Lines 226-229: We have included age in the matrix correlation [It is unclear whether there was any correlation of symptoms with participants’ age]
  • Lines 396-400: We have included in the limitations of the work, that the inattention symptoms could be more associated with anxiety as authors like Bloemsma have indicated. [Students with more inattention show lower levels of emotional, social and physical self-concept mediated by anxiety. It should be addressed whether anxiety levels were however higher in students with more inattention. If this is not the case, the issue should be discussed as well.]
  • Lines 304-305. We have taken into account the symptomatology as a continuum in the discussion. Line 347: we have avoided using harsh conclusions [When drawing their conclusions, the Authors should pay more attention and discussion to the continuum of symptomatology in patients with ADHD. To assert that “in the case of hyperactivity-impulsivity there is no association between this set of symptoms and students’ self-concept” is a hazard in the opinion of this reviewer].

Reviewer 2 Report

Dear Authors, your paper is interesting. It is clear and the research design was appropriate. Results are clearly presented. The main question question addressed by the research is very interesting even if it is in the area of symptomatology and not of disorder tout-court. However, this limit is reported in the paper. I appreciated the topic even if it is not very original. Your study adds information in terms of wide phenotype rather that in terms of disorder well diagnosed. The text is very clear to read, the methodology is well described. The paragraph of the conclusions is in line with the arguments presented. In general the authors address the main question of their reseaerch project. I consider this study as acceptable in his current form.

Author Response

Thank the reviewer for the kindly comments and the analysis of our work